# GRADUAL DOMAIN ADAPTATION IN THE WILD: WHEN INTERMEDIATE DISTRIBUTIONS ARE ABSENT

## ABSTRACT

We focus on the problem of domain adaptation when the goal is shifting the model towards the target distribution rather than learning domain invariant representations. It is shown that under the following two assumptions: (i) access to samples from intermediate distributions, and (ii) samples being annotated with the amount of change from the source distribution, self-training can be successfully applied on gradually shifted samples to adapt the model toward the target distribution. We hypothesize having (i) is enough to enable iterative self-training to slowly adapt the model to the target distribution by making use of an implicit curriculum. In the case where (i) does not hold, we observe that iterative self-training falls short. We propose GIFT (Gradual Interpolation of Features toward Target), a method that creates virtual samples from intermediate distributions by interpolating representations of examples from source and target domains. Our analysis of various synthetic distribution shifts shows that in the presence of (i) iterative self-training naturally forms a curriculum of samples which helps the model to adapt better to the target domain. Furthermore, we show that when (i) does not hold, more iterations hurt the performance of self-training, and in these settings, GIFT is advantageous. Additionally, we evaluate self-training, iterative self-training and GIFT on two benchmarks with different types of natural distribution shifts and show that when applied on top of other domain adaptation methods, GIFT improves the performance of the model on the target dataset.

## 1 INTRODUCTION

Learning algorithms are notorious for not being robust to changes in the environment, and their performance often drops when there is a shift in the data distribution (Taori et al., 2020; Hendrycks et al., 2021; Koh et al., 2021). To be robust to the changes in the distribution, learning algorithms either need to learn representations that are invariant to the shift, or they should update their parameters to be more aligned with the new distribution. In either setting, we are still far from a satisfactory solution. While unsupervised domain adaptation techniques commonly rely on learning domain invariant representations, our focus is on shifting the model towards the target distribution. We consider the unsupervised setting where we do not have access to labels on the target domain. The setting we consider is similar to the setting suggested by Wang et al. (2021).

We look at this problem through the lens of curriculum learning. Curriculum learning (Elman, 1993; Sanger, 1994; Bengio et al., 2009) suggests presenting easier samples early on in the training process and gradually increasing the difficulty. In unsupervised domain adaptation, this is equivalent to gradually changing the distribution from the source domain to the target domain, i.e., getting the model to adapt to intermediate distributions before being exposed to the target domain. Kumar et al. (2020) show that if learning algorithms are exposed to gradual changes in the data distribution under a self-training regime, the generalization gap (from source to target distribution) will be much lower. However, the problem remains unresolved when we do not have access to the intermediate steps of the distribution shift. Findings from Kumar et al. (2020) suggest that to be able to adapt the model by self-training, two conditions should be satisfied: (i) Access to samples from intermediate distributions between source and target; (ii) Access to information about the amount of shift for each sample. While (ii) is hard to achieve in practice, (i) may be the case in many real-world scenarios. We hypothesize that if (i) holds, i.e., the target distribution is diverse enough to include samples from intermediate distributions; iterative self-training (applying self-training iteratively) and filtering examples based

on the confidence of the model, incorporates an implicit curriculum and this curriculum helps the model to gradually adapt to the target domain. Furthermore, for cases where (i) does not hold, we propose GIFT (Gradual Interpolation of Features toward Target). GIFT creates virtual examples from intermediate distributions by linearly interpolating between source and target data in the input and feature space of a neural network. Note that such interpolations are shown to be meaningful in works that established the idea of mixing up representations (Verma et al., 2019a; Mroueh, 2020a). To gradually increase the difficulty of the virtual samples, we change the linear interpolation coefficient such that the samples start at the source distribution and gradually move towards the target distribution during training. Figure 1, demonstrates how GIFT can improve iterative self-training, in an example with a Two-moon dataset (two interleaving half circles).

Our focus in this paper is not to provide a new state of the art on a specific benchmark, as we are limited by the availability of benchmarks that are specifically designed for gradual domain adaptation. Instead, we intend to provide a proof of concept for (1) how, why and when iteratively applying self-training is helpful for gradual domain adaptation (Section 4.1 and 4.2); (2) There are cases that not only iterative self-training does not help the model but also it hurts model performance on both source and target domains. In those cases, GIFT can mitigate the issue by using interpolations of features as intermediate examples (Section 4.1). We evaluate iterative self-training and GIFT for unsupervised domain adaptation on datasets with natural distribution shifts and show that they improve the performance of models on the target dataset compared to non-iterative self-training. On a synthetic benchmark, we show that in the absence of (i), GIFT performs better than iterative self-training. By tracking the accuracy and confidence of a model on different subsets of the target distribution, we show that with both GIFT and iterative self-training the order in which the models learn to correctly classify examples is aligned with their degree of perturbation. GIFT has two advantages over iterative self-training: (1) It works better when the number of training iterations is limited, (2) it works better when the target distribution is not diverse enough to include a mixture of easy and hard examples.

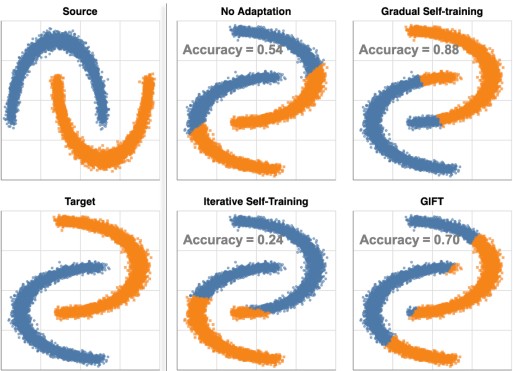

Figure 1: Demonstrating the power of self-training for shifting the model on the Two-moon dataset. On the left side, different colors determine the two ground truth classes. On the right, different colors depict predicted classes. Here the target data (bottom left) is the ($90°$) rotated version of the source data (top right). In this example, it is not possible to achieve a good performance on both source and target at the same time, since they have conflicting labels for similar inputs. While the performance of the model trained only on source is around $\sim 50\%$ on the target data, if we have access to ground truth intermediate steps we can improve this number by $\sim 40\%$ by gradual self-training (Kumar et al., 2010). In the absence of samples from intermediate distributions, our proposed method(GIFT) can increase the performance by $\sim 20\%$ compared to no adaptation, while iterative self-training without access to intermediate examples leads to even worse accuracy. For this example, we use an MLP with one hidden layer and a Relu activation function. The code is provided at `https://github.com/anonymouse-github/GIFT`.

## 2 SELF-TRAINING FOR UNSUPERVISED DOMAIN ADAPTATION

Being able to properly handle distribution shift is one of the primary concerns of machine learning algorithms. A common setting for this problem is unsupervised domain adaptation, where we have

access to labeled data from one or multiple source domains, and unlabeled data from the target domain. Let $(X^s, Y^s)$ denote the labeled data in the source domain, where $X^s \in \mathbb{R}^{n_s \times d}$ is the sample input matrix and $Y^s \in \mathbb{R}^{n_s \times k}$ is the corresponding label matrix. Let $X^t \in \mathbb{R}^{n_t \times d}$ denote the unlabeled target domain data. Here $n_s$ and $n_t$ determine number of examples from source and target domains respectively, and $k$ determines the number of classes. While, it is not necessary for simplicity we assume number of classes and generally the task is the same in source and target. We assume some underlying or common features exist between source and target, while there is a substantial distribution shift between the two domains. The goal is to bridge the domain difference and learn a classifier minimizing the empirical risk on the target domain. The first assumption is needed for domain adaptation to be successful (Ben-David et al., 2008). The latter emphasizes that a model trained on the source domain presents a noticeable performance gap on the target domain and hence needs to adapt to the target distribution. Self-training uses a teacher model trained on source domain, $P_s$, to produce pseudo labels, along with confidence scores, on unlabeled data from the target domain, $P_t$, and uses these predictions to train a student model. In iterative self-training, the teacher model is periodically updated and replaced by the student model. We examine how, in iterative self-training, the confidence scores of the teacher model guide the student model to adapt to the target domain, and whether we can devise supplementary schemes that help the student adapt better.

## 3 GIFT: Self-training with Gradual Interpolations

If in addition to labeled data from the source domain and unlabeled data from the target domain, we also have access to the intermediate distributions, i.e., unlabeled examples from data distributions between source and target, we can use them to boost the performance of the model on the target domain, by applying self-training in a gradual manner. Kumar et al. (2020) show that this gradual adaptation leads to a lower error bound on the target domain. In order for this approach to be applicable, we need samples from intermediate steps that are annotated with the amount of shift from the source distribution. However, in many cases in practice, we either do not have access to samples of intermediate distributions or such examples do not exist. Even if they exist, it is less likely that they are annotated with the degree of shift.

To circumvent the issue of lack of samples from intermediate distributions, we create the virtual samples from intermediate distributions by interpolating the input and hidden representations of the data from the source domain $P_s$ and target domain $P_t$. Namely, let $M_\phi$ be a neural network model trained on source data and let $z_i^s$ correspond to the representation of input $x_i^s \in X^s$ from the source domain. We choose $x_j^t$ as a sample from the target domain (we explain the procedure to pick $x_j^t \in X^t$ below). Let $z_j^s$ correspond to the representation of input $x_j^t \in X^t$ that can be sampled from any of the layers of a neural network (including the input layer), and $\lambda \in [0, 1]$. We generate

$$\widehat{z}_{ij} = (1 - \lambda)z_i^s + \lambda z_j^t, \qquad \lambda \in [0, 1] \tag{1}$$

as a sample representation of a virtual intermediate distribution. Note that we do not seek to find a corresponding input to the interpolated representation. Gradually increasing $\lambda$ from 0 to 1 corresponds to feature representations that change from corresponding source distribution representations ($\lambda = 0$) to the target distribution representation ($\lambda = 1$). **lambda_scheduler** in algorithm 1, represent the scheduling function for gradually increasing $\lambda$, which can be anything, in our experiments we use a linear function with fixed step size, $\delta$, where the value of lambda increases by a constant value after every $\delta$ training steps.

To choose which example from the source is interpolated with which example from the target, we either randomly align the samples or apply a cost-based alignment method based on the $L_2$ distance of representations, and the similarity/equality of labels on the source and predicted target labels. To apply a cost-based alignment method, we use the Sinkhorn matching algorithm (Sinkhorn, 1966; Peyré & Cuturi, 2019) to approximate the alignment with the lowest cost. Although iterative cost-based alignment methods such as Sinkhorn come at a computational cost, compared to random alignment, we have observed that random alignment without taking (pseudo) labels into account leads to worse performance. To find a better trade-off between alignment cost and performance, in our experiments, we also tried a non-iterative heuristic (pseudo) label-based random alignment, where we randomly align examples that have the same (pseudo) labels. This is shown in Algorithm 2. In our experiments, we report the result from cost-based alignment. We observed that pseudo-random alignment and cost-based alignment lead to more or less similar results and in practice, we could simply use the

---

**Algorithm 1** GIFT: Gradual Interpolation

$P_s,$ : source domain; $P_t$: target domain
$\mathcal{M}$: Neural net (maps input to predictions).
$\mathcal{M}^{:L}$: partial neural net (maps input to features of layer L)
$\mathcal{M}^{L:}$: partial neural net (maps features of layer L to predictions)
$\phi$: student neural net parameters; $\theta$: teacher neural net parameters
$\delta$: step size for interpolation coefficient $\lambda$.
$N$: number of training iterations for each teacher update.
$m$: number of examples in a batch.
$\alpha$: confidence threshold
Notation: $[m] = \{1, ...m\}$.

1:  $\mathcal{M}_\phi \leftarrow \mathcal{M}_\theta, \lambda \leftarrow 0$
2:  **while** $\lambda <= 1$ **do**
3:     **for** $step \in [N]$ **do**
4:        $(x^s_{[m]}, y^s_{[m]}) \sim P_s$
5:        $x^t_{[m]} \sim P_t$
6:        $z^s_{[m]} \leftarrow \mathcal{M}^{:L}_\phi(x^s_{[m]})$
7:        $z^t_{[m]} \leftarrow \mathcal{M}^{:L}_\theta(x^t_{[m]})$
8:        $y^t_{[m]} \leftarrow \mathcal{M}_\theta(x^t_{[m]})$
9:        $\text{index}_s, \text{index}_t \leftarrow \textbf{align}(y^s_{[m]}, y^t_{[m]})$ (Algorithm 2)
10:      $\widehat{z}_{[m]} \leftarrow (1 - \lambda) \times z^s_{\text{index}_s} + \lambda \times z^t_{\text{index}_t}$
11:      $\widehat{y}_{[m]} \leftarrow \mathcal{M}^{L:}_\theta(\widehat{z}_{[m]})$
12:      $\text{conf\_ranks} \leftarrow rank(max(\widehat{y}_{[m]}) - min(\widehat{y}_{\{...\}}))$
13:      $\text{conf\_indices} \leftarrow \text{conf\_ranks}[: \alpha]$
14:      $\widehat{z}_{[m]}, \widehat{y}_{[m]} \leftarrow \widehat{z}[\text{conf\_indices}], \widehat{y}[\text{conf\_indices}]$
15:      Fit $\mathcal{M}_\phi$ to $(\widehat{z}_{[m]}, \widehat{y}_{[m]})$
16:    **end for**
17:    $\lambda \leftarrow \textbf{lambda\_scheduler}(\lambda, \delta)$
18:    $\mathcal{M}_\theta \leftarrow \mathcal{M}_\phi$
19: **end while**

---

**Algorithm 2** Align: Label-based Random Alignment

1:  **Input:** $y^s_{[m]}, \hat{y}^t_{[m]}$
2:  **Output:** $\text{index}_s, \text{index}_t$
3:  $\text{index}_s \leftarrow [1, 2, ..., len(y^s_{[m]})]$
4:  $\text{index}_t \leftarrow []$
5:  **for** $i \in \text{index}_s$ **do**
6:     $\text{indices} \leftarrow [1, 2, ..., len(\hat{y}^t_{[m]})]$
7:     $\text{shuffled\_indices} \leftarrow permute(\text{indices})$
8:     $\text{matching\_scores}[m] \leftarrow \begin{cases} 1 & \text{for } \hat{y}^t[m] = y^s[i] \\ 0 & \text{for } \hat{y}^t[m] \neq y^s[i] \end{cases}$
9:     $\text{index} \leftarrow argmax(matching\_scores[\text{shuffled\_indices}])$
10:    $\text{index}_t.append(\text{shuffled\_indices}[\text{index}])$
11: **end for**

---

pseudo-random alignments. It is important to note that both for cost-based and pseudo-random alignment we use the pseudo labels predicted by the teacher model since we do not have access to ground truth labels for data points from the target domain.

In standard iterative self-training (Habrard et al., 2013), the data used in each self-training iteration is a subset of the target distribution, whereas in GIFT we apply the iterative self-training procedure on virtual intermediate distribution representations to gradually adapt the model to the target domain. We

start by using a teacher model trained on the source distribution and move to train the student model in the representation space. The self-training procedure at each step proceeds by assigning pseudo labels to virtual intermediate representations that are generated by equation 1. Next, the student model is updated using pseudo-labeled representations. Then the student becomes the teacher for the next iteration and the procedure continues. We start with $\lambda = 0$, which generates representations from the source domain. At each iteration of self-training, we increase the value of $\lambda$ to generate virtual representations that are closer to the target distribution and hence gradually move the model toward the target distribution ($\lambda = 1$). The details are shown in Algorithm 1. Similar to standard iterative self-training, we assign a confidence score to each pseudo-labeled data point and only update the student model with pseudo-labeled data which their confidence score is in the top $q$-percentile. In Alorithm 1, $q$ is specified by *Confidence Thershold*. As a confidence score, we use the gap between the highest and lowest logit for each sample, as proposed in Kumar et al. (2020).

## 4 EXPERIMENTS

In our experiments, first, we examine the power of iterative self-training and GIFT for unsupervised domain adaptation on a dataset with synthetic shifts, where we can track the amount of shift and control the gap between the source and targets distribution. We demonstrate the implicit curriculum followed by iterative self-training and GIFT and show that when condition (i) does not hold, GIFT performs better than iterative self-training. Next, we showcase the performance of these methods in more realistic settings, on datasets with natural distribution shifts.

The training setup that we propose has three main phases: (i) Pre-training: Pre-training the model on a large scale dataset (ImageNet-1k in our experiments) (ii) Fine-tuning: Training the pre-trained model on a labeled source domain with or without leveraging samples from an unlabeled target domain. (iii) Adaptation: Applying a self-training based approach to shift the model towards the unlabeled target domain. In the fine-tuning stage, step (ii), we compare (A) simple fine-tuning on the source domain with standard augmentation techniques, i.e., random flip and random crop, (B) fine-tuning on the source domain with variants of Mixup (Zhang et al., 2018a) as an augmentation technique, and (c) Domain Adversarial training that uses unlabeled examples from the target domain to learn domain invariant representations (Ganin et al., 2016). In step (iii), the adaptation phase, we compare three self-training based strategies: (1) one-step self-training, (2) iterative self-training and (3) GIFT, which is similar to iterative self-training except that during the intermediate iterations the model is trained with samples from virtual intermediate states instead of actual samples from the target distribution.

### 4.1 BENCHMARKS WITH SYNTHETIC PERTURBATIONS

To investigate the effect of the type and degree of shift, on the success of iterative self-training and GIFT we compare their performance, on a synthetic benchmark. In this benchmark, the target domain is created by applying synthetic perturbations on examples from the source domain, such as applying noise, rotating, scaling or translating images.

To create these benchmarks, we split the training set of CIFAR10 (Krizhevsky et al., 2009) into equal-sized splits, where each split contains examples with different degrees of perturbation. We use four types of perturbation: rotation, scale, translate and blur. For rotation, we split the training data into three parts. In the first split, images have a rotation angle of 0 to 5 degrees. In the second split, images have a rotation angle of 5 to 55 degrees, and in the last split, images have a rotation angle of 55 to 60. We use the first split, 0_5, as the source domain and the third split 55_60 as the target domain. For scale, translate and blur, we split the training data into two splits, where we use the first part with no perturbation as the source domain, and we apply the perturbation on the second part to get the out-of-distribution target domain. For both scale and translate, we have no variation in the source and some variation in the target. For the blur, we only have one degree of blurring perturbation in the target and none in the source domain.

The results on CIFAR10 with different perturbations are shown in Tables 1 and 2. They indicate the superiority of GIFT to iterative self-training. The advantage of GIFT over iterative self-training is more apparent when the target distribution does not include a diverse set of samples (Translated vs Blur) or when the two distributions are not overlapping (Translated(0%-100%) vs Translated(50%-100%)). Additionally, compared to GIFT, the performance of iterative self-training drops more

Table 1: Results of different adaptation techniques on perturbations of CIFAR10 in terms of accuracy on the test set, with WideResnet18-10. The total number of training steps is 1000, with a batch size of 512. The number of self-training iterations is 5 and 20 for iterative self-training, and GIFT respectively. None refers to the zero-shot performance of the pretrained model. In all cases, GIFT outperforms all the baselines.

| TARGET DOMAIN | NONE | SELF-TRAINING | ITERATIVE SELF-TRAINING | GIFT |
|---|---|---|---|---|
| ROTATED CIFAR10 | 0.38 | 0.406 | 0.396 | **0.436** |
| SCALED CIFAR10 | 0.559 | 0.558 | 0.578 | **0.615** |
| TRANSLATED(0%-100%) CIFAR10 | 0.551 | 0.676 | 0.808 | **0.859** |
| TRANSLATED(50%-100%) CIFAR10 | 0.262 | 0.421 | 0.658 | **0.729** |
| BLURRED CIFAR10 | 0.351 | 0.355 | 0.311 | **0.545** |

Table 2: Performance of GIFT and iterative self-training for WideResnet18-10 trained on perturbations of CIFAR10 in terms of accuracy on the target. The total number of training steps is 500, with a batch size of 512. The number of teacher updates is 2 and 20 for iterative self-training and GIFT, respectively. Comparing these results to Table 1 with the total number of training steps of 1000 indicates a noticeable drop in the performance of iterative self-training while GIFT's performance is more robust.

| TARGET DOMAIN | ITERATIVE SELF-TRAINING | GIFT |
|---|---|---|
| ROTATED CIFAR10 | 0.408 | **0.4176** |
| SCALED CIFAR10 | 0.573 | **0.6767** |
| TRANSLATED(0%-100%) CIFAR10 | 0.629 | **0.8349** |
| TRANSLATED(50%-100%) CIFAR10 | 0.477 | **0.8319** |
| BLURRED CIFAR10 | 0.3551 | **0.5213** |

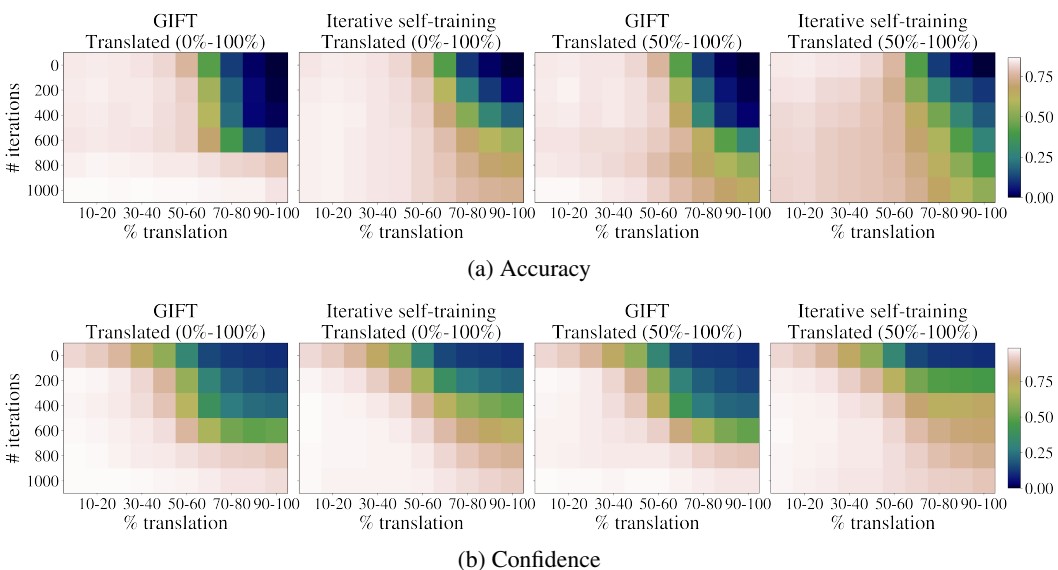

(a) Accuracy

(b) Confidence

Figure 2: Accuracy and Confidence of GIFT and iterative self-training, as a function of the number of training iterations on different bins of the translated CIFAR10. We evaluate the accuracy for the test data with translations between 0 and 100% (where zero means no shift, and 100% means the maximum possible amount of shift, i.e., IMAGE WIDTH/2). The iterative self-training model has 5 teacher updates over 1000 training iterations (i.e. updates happen after 0, 200, 400, 600, 800 iterations). GIFT has 20 teacher updates over 1000 iterations (i.e. updates happen after 0, 50, 100, ..., 950 iterations). The left two panels in each row correspond to models trained for the target dataset Translated (0%-100%) CIFAR10. The right two panels in each row correspond to models trained for the target dataset Translated (50%-100%) CIFAR10.

significantly when decreasing the total number of training steps. We can see this by contrasting the results in Table 1 and 2.

**Investigating the Curricula:** To empirically confirm the hypothesis that both iterative self-training and GIFT gradually guide the model to fit the out-of-distribution target distribution, we track the accuracy and confidence of the models on different subsets of the target data throughout the training.

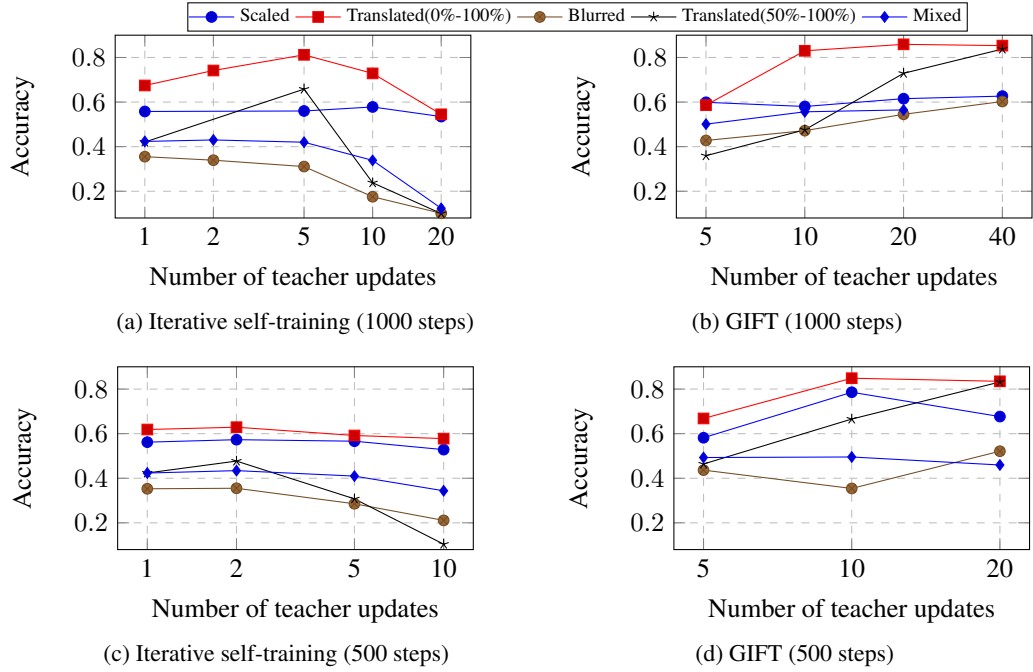

Figure 3: Accuracy of iterative self-training and GIFT on perturbations of CIFAR10 as a function of the number of teacher updates when the total number of training steps is 1000 and 500. Accuracies of both models improve by increasing the number of self-training iterations up to a threshold. Beyond this threshold, while iterative self-training performance deteriorates, GIFT saturates and hence shows more robustness.

This is shown in in Figures 2. We see that, for both methods, the accuracy and confidence measures are increasing incrementally from easier examples, i.e., examples with smaller amounts of perturbation, to harder examples, i.e., examples with larger amounts of perturbation.

For GIFT, the number of iterations is tied to the steps in which we increment the interpolation coefficient, $\lambda$, and it is interesting to see that there is a correlation between $\lambda$ and the accuracy and confidence of the model on actual intermediate steps, as shown in Figure 2. For iterative self-training, the model selects training examples for which to compute the loss based on its confidence. Hence, tracking the confidence as a function of translation percentage during training allows us to see whether the model is indeed selecting less perturbed examples earlier and more perturbed examples later, thus creates its own curriculum. As we see in Figure 2, for both iterative self-training and GIFT the confidence of the models decreases gradually for data with an increased level of perturbation. Note also that the confidence scores show very similar patterns to the accuracies. This confirms that the reason behind the success of iterative self-training is the implicit curriculum strategy, and that the gradual interpolation strategy that we employ in GIFT can be a good proxy for gradual self-training when we do not have access to the gradually shifted data.

**Effect of number of intermediate steps:** We examine the effect of the number of intermediate steps when the total number of training steps is fixed. As illustrated in Figure 3, increasing the number of intermediate steps (teacher updates), which means taking smaller steps in the gradual adaptation procedure, leads to a better performance up to a threshold for GIFT and iterative self-training. If we keep increasing the number of intermediate steps, the performance of the models decreases rapidly for iterative self-training. Whereas GIFT is more robust to the number of intermediate steps. This decrease in the performance is potentially due to accumulations of the errors of the self-training process or because with a fixed number of training steps, increasing the number of intermediate steps leads to a decrease in the number of iterations in each self-training step, which could mean the model can not adapt well to each intermediate step. The robustness of the GIFT, in this case, could mean that this method needs fewer iterations in each intermediate step. Comparing Figure 3, we see that for both iterative self-training and GIFT, decreasing the number of total training steps (500 vs 1000), reduces the effective number of teacher updates. This confirms the hypothesis that the model can

Table 3: Accuracy on target domain on benchmarks with natural distribution shift. For the experiments in this table we use a ResNet-101 pretrained on ImageNet-1k.

| Method | FMoW | Camelyon17 |
|---|---|---|
| Fine-tuned on Source (A) | 0.502 | 0.807 |
| Mixup-Convex (B) | **0.525** | 0.591 |
| Mixup-Wasserstein (C) | 0.499 | 0.896 |
| DANN (D) | 0.505 | **0.934** |
| Best (A, B, C, D) + Self-training | 0.530 | 0.962 |
| Best (A, B, C, D) + Iterative Self-training | **0.539** | 0.966 |
| Best (A, B, C, D) + GIFT | **0.539** | **0.973** |

benefit from more teacher updates if it has enough time to properly adapt to each intermediate step. We provide more analysis for this in Appendix C.

## 4.2 BENCHMARKS WITH NATURAL DISTRIBUTION SHIFT

We report results on two datasets with natural distribution shifts: FMoW (Christie et al., 2018) and Camelyon17 (Bandi et al., 2018) from the WILDS benchmark (Koh et al., 2021). Here we briefly introduce each of these datasets:

**FMoW** is a variant of the Functional Map of the World dataset that contains satellite imagery of the earth. The images belong to 62 building or land use categories, and the domain represents both the year the image was taken as well as its geographical region. Here we only address the domain shift problem over time. For the adaptation phase, we use unlabeled samples from the out-of-distribution test split of the dataset.

**Camelyon17** is the patch-based variant of Camelyon17 (Bandi et al., 2018) with hospitals $(0, 1, 2, 3, 4)$ as domains. The task is to predict if a given region of tissue contains a tumor. We use $0, 1, 2$ as source during fine-tuning, but for adaptation we only use labeled data from domain $0$. We use $3$ as the target domain. This is an example of a dataset, for which annotated intermediate distributions between different domains do not exist naturally.

As shown in Table 3 on datasets with natural shift, all three self-training based approaches, self-training, iterative self-training and GIFT improve the accuracy over the best fine-tuned models. While on FMoW performance of GIFT is on par with iterative self-training, it achieves a higher accuracy on Camelyon17. An advantage for GIFT over iterative self-training in these scenarios is that, the performance of GIFT does not rely on the assumption (i) ,that the target distribution should include samples from intermediate steps since in the intermediate steps of training it merely learns from the interpolated samples.

## 5 RELATED WORK

**Unsupervised domain adaptation:** In unsupervised domain adaptation, we have access to labeled examples from the source domain(s) and unlabeled examples from the target domain(s) and the goal is to get a good performance on the target domain. Unsupervised domain adaptation techniques fall within three main categories (Sun et al., 2020): (1) Methods based on matching the feature distributions of source and target domains. Algorithms in this group rely on the assumption that the models can learn domain invariant representations, and they employ different self-supervised based losses to enforce this invariance by exposing the model to the unlabeled data from the target distribution (Ganin et al., 2016; Ben-David et al., 2006). (2) Methods based on transforming source and target distributions. They analyze the input space and project source and target data to a lower dimensional manifold and try to find a transformation between the two (Fernando et al., 2013; Gopalan et al., 2011; Harel & Mannor, 2010). Another approach in this line of work is to transform the source data to be as close to the target data as possible. For example, Sun et al. (2017) matches the second order statistics of the input spaces. (3) Self-training based methods.

**Self-training:** Recent works have shown significant progress using self-training in computer vision (Xie et al., 2020; Yalniz et al., 2019; Zoph et al., 2020). Self-training has also been used for domain adaptation by generating pseudo labels in the target domain and directly training a model for the target domain (Xie et al., 2018; Saito et al., 2017; Chang et al., 2019; Manders et al., 2018; Zou et al.,

2019; 2018). Xie et al. (2018) align labeled source centroids and pseudo-labeled target centroids. (Chang et al., 2019) uses different normalization parameters for source examples and pseudo labeled examples in the target domain. Zou et al. (2019) introduced label-regularized self-training which generates soft pseudo-labels for self-training. Different from these works, we use a curriculum learning approach where we generate pseudo labels for intermediate virtual examples and gradually adapt the model to the target domain. **Curriculum learning:** Curriculum learning (Elman, 1993; Sanger, 1994; Bengio et al., 2009) has led to better performance in terms of generalization and/or convergence speed in many domains such as computer vision (Pentina et al., 2015; Sarafianos et al., 2017; Guo et al., 2018; Wang et al., 2019), natural language processing (Cirik et al., 2016; Platanios et al., 2019) and neural evolutionary computing (Zaremba & Sutskever, 2014). On the other hand, there have been some negative results in neural machine translation (Kocmi & Bojar, 2017; Zhang et al., 2018b; 2019). In this work, we investigate it in the unsupervised domain adaptation scenario. Different notions of "difficulty" of examples are used in the literature, such as using the loss value of a pre-trained model (Bengio et al., 2009), or the first iteration in which an example is learned and remains learned after that (Toneva et al., 2019). Jiang et al. (2020) have proposed using a consistency score calculated based on the consistency of a model in correctly predicting a particular example's label trained on i.i.d. draws from the training set. Wu et al. (2021) show that all these difficulty scores are consistent. Here, we use the coefficient of linear interpolation between source and target representation as a measure of the difficulty of a (virtual) sample.

**Leveraging synthetic data for domain adaptation:** We build on top of the existing work that incorporate synthetic data to bridge between source and target domain Gopalan et al. (2014); Gong et al. (2019); Cui et al. (2020); Na et al. (2021). Similar to Na et al. (2021) we use mixup to create intermediate samples, but instead of using fixed mixup rates, we incorporate an annealing strategy so the created instances, gradually move away from the source domain and get closer to the samples from the target domain. Additionally, we apply mixup both in the input layer and representations obtained from various layers of the models. Gong et al. (2019) proposes to create intermediate domains using CycleGAN (Hoffman et al., 2018), wheras in GIFT, we create the intermediate domains in the representational space of the model and abstract away from actual images corresponding to those representations.

## 6    DISCUSSION AND CONCLUSION

We establish the importance of having a proper curriculum for the success of self-training algorithms for domain adaptation. We demonstrate that iterative self-training can successfully adapt the model to the new target distribution if the target distribution contains the intermediate examples. We show that iterative self-training is less effective if the source and target distributions are not overlapping. Our proposed method, GIFT, is specifically designed to deal with cases where there is a gap between source and target distributions. In such scenarios, applying self-training iteratively, and filtering examples based on the confidence of the model would not result in a proper curriculum since the intermediate examples are missing. We propose and perform controlled experiments on distribution shift on a synthetic benchmark created from CIFAR10. More specifically, we control the amount of shift, the type of shift, and whether there is a gap between the data distributions in the source and target domain. Our experiments confirm the importance of the presence of samples from intermediate distributions for the success of iterative self-training and show that in the absence of intermediate samples, GIFT improves the results over iterative self-training. We observe similar results on datasets with natural distribution shifts.

GIFT is inspired by existing works that rely on interpolating representations in the input or feature space (Gong et al., 2012; Verma et al., 2019a;b; Neyshabur et al., 2020). An interesting direction for future work is to investigate other ways of interpolating in the feature space and explore the use of more sophisticated interpolation schemes, such as schemes based on optimal transport. Moreover, the experiments in this paper are limited to image classification. Extending our approach beyond classification and to other data modalities such as textual data is an interesting next step. Lastly, we emphasise the importance of designing benchmarks specifically for the purpose of evaluating gradual domain adaptation techniques under different conditions, as one of the most important steps in this line of research.

REPRODUCIBILITY STATEMENT

We have shared the details of the proposed algorithms and experimental setup in sections 3 and A. Additionally, we provide a simple implementation of the main algorithms proposed at `https://github.com/anonymouse-github/GIFT`.

We ran experiments on a dataset with synthetic shifts, for which we thoroughly explain how it is created in section 4.1, as well as two datasets with natural distribution shifts which are already available as part of the WILDS benchmark (Koh et al., 2021).

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

## A   EXPERIMENTAL SETUP

**Optimizer and learning rate schedule.**  In all our experiments on datasets with natural distribution shift, we use Adam optimizer. When training on the labeled source domain we use a learning rate schedule with cosine decay and initial learning rate of 1e-4 and when adapting to the target domain we use a learning rate of 1e-5 with an exponential decay rate of 0.9. For pretraining the models on ImageNet-1k, we use a batch size of 1024 and the learning rate schedule is linear warmup (for 5 epochs) + cosine decay with the base learning rate of 0.1.

For Perturbed Cifar10 experiments, we use SGD with momentum. During the pretraining stage on the source domain, the learning rate schedule is cosine decay with an initial learning rate of 0.1. In all experiments, we use L2 loss with the weight of 1e-5. During the adaptation phase, we use a batch size of 256 and the learning rate is constant and set to 1e-3. For experiments on Perturbed Cifar10 with a higher number of training steps (20000 steps), we use a lower learning rate of 1e-4 in the adaptation phase.

**Neural network architectures.**  For experiments on datasets with natural shifts, we use a ResNet101. For the experiments on Perturbed Cifar10, we use a WideResNet28-10 with a dropout rate of 0.3. For the virtual interpolations in GIFT: we use the first three layers (input, initial convolution layer, and the layer above it).

**Training on source during the adaptation phase.**  We do not train the model on the source data during the adaptation phase. While in some cases this could result in a better performance on both the source and target domain, our assumption here is that there is no reason for the source and target data to be compatible, i.e. it is possible for the model to not be able to fit both distributions simultaneously.

**Regularization**  During adaptation, we use the weight decay of 0.01. In addition to weight decay, we use another regularization term that encourages the model to stay close to its initial state. This regularization term is simply computed as the L2 distance of the parameters of the model and their value at its initial state. We set the weight for this factor to 0.001 in all adaptation experiments.

**Backpropagating gradients through interpolations.**  We train the model with the representations of virtual examples that we create by interpolating the representations from real examples. During pretraining on the source domain, we use manifold mixup regularization, where we interpolate between representations of labeled source examples. During the adaptation stage that is part of GIFT, we interpolate between labeled source representations and unlabeled target representations. One important hyper-parameter related to this is whether in the backward pass we back-propagate the gradients all the way down to the input layer or stop the gradients at the layer in which the interpolated representations are computed. In our experiments, similar to Verma et al. (2019a) we allow the gradients to pass through the interpolated activations.

## B   TRAINING ON LABELED SOURCE DOMAIN

We compare four different approaches for training the model on labeled source data.

**Standard fine-tuning:**  Given labeled examples from a source domain, we employ a model that is pretrained on some large scale dataset, Imagenet-1k in our case, replace its head (projection layer) with a new head for the task at hand, and update all its parameters to fit the source domain data. **Mixup with convex combination interpolations:**  During fine-tuning on labeled source data, we apply mixup/manifold (Zhang et al., 2018a; Verma et al., 2019a) on the input and activation from the first layer of the model, and to compute the interpolations we simply compute the convex combination of features for two randomly aligned examples. **Mixup with wasserstein interpolations:**  During fine-tuning on labeled source data, we apply a variant of mixup/manifold mixup where interpolations are computed using the closed-form Monge Map for Gaussian Wasserstein distances. In our experiments, we observed that in some cases, interpolating examples in this alternative way leads to better results compared to the convex interpolations used in Verma et al. (2019a). **Domain adversarial neural networks (DANN):**  Given labeled samples from a source domain and unlabeled samples from the target domain, DANN (Ganin et al., 2016) learns domain invariant representations, while minimizing its error on labels source data. In our experiments, the output of the prelogits layer is fed to the domain classifier, and the scheduling of the weight of the domain classification loss is the same as what is suggested in Ganin et al. (2016).

### B.1 MANIFOLD MIXUP WITH WASSERSTEIN INTERPOLATION

During training on the labeled source dataset, we use a variant of manifold mixup (Verma et al., 2019a) with an adapted strategy for interpolation. In manifold mixup (Verma et al., 2019a), representations of virtual examples are created by linearly interpolating representations of two randomly aligned examples $(x_i, x_j)$ in a randomly selected layer $L$ of the neural network $M_\theta$. The labels for the interpolated examples, $\hat{y}_{ij}$, are computed by interpolating the labels of the aligned examples, $(y_i, y_j)$, using the same interpolation coefficient, $\lambda$. This is summarized in equation 2.

$$\begin{aligned} \hat{z}_{ij} &= (1 - \lambda)\mathcal{M}_\theta^{:L}(x_i) + \lambda\mathcal{M}_\theta^{:L}(x_j) \\ \hat{y}_{ij} &= (1 - \lambda)y_i + \lambda y_j \end{aligned}. \tag{2}$$

Here $M_\theta^L$ denotes the part of the neural network that outputs the activations of layer $L$. The interpolated representations are then fed into the rest of the neural network at layer, $L$, and together with the interpolated labels, they serve as additional 'data' to which the model is fit.

In our experiments, we take a different approach to interpolation. Inspired by the style transfer method discussed in Mroueh (2020b), we use interpolations based on the Wasserstein distance between two Gaussian distributions that are fit to representations of two input images. i.e., the spatial features in the representations of two images are the data points for two datasets. We estimate the empirical mean and diagonal covariance matrices for these data points and use the closed-form optimal transport map between two Gaussian distributions to interpolate the spatial features between two representations. More precisely, given two images $x_i$ and $x_j$, we compute representations $z_i = M_\theta^L(x_i)$ and $z_j = M_\theta^L(x_j)$, where $z_i$ and $z_j$ are three-dimensional tensors with a width $W^L$, height $H^L$ and channel size $C^L$. Each spatial feature vector of size $C^L$ within $z_i$ and $z_j$ is considered one data point. We compute the average feature vectors within $z_i$ and $z_j$, denoted by $\mu_i$ and $\mu_j$ respectively, as well as the variances $\sigma_i^2$ and $\sigma_j^2$. Note that we are approximating the empirical covariance matrices with diagonal matrices with the variances $\sigma_i^2$ and $\sigma_j^2$ on the diagonals. Given these quantities, we can compute the closed-form Monge map between two Gaussian distributions with diagonal covariance matrices as

$$T_{z_i \to z_j}(z) = \mu_j + \operatorname{diag}\left(\frac{\sigma_j}{\sigma_i}\right)(z - \mu_i). \tag{3}$$

Here $z$ is understood to be a feature map of the same size as $z_i$ and $z_j$. Interpolated representations are then computed with

$$\hat{z}_{ij} = (1 - \lambda)z_i + \lambda T_{z_i \to z_j}(z_i). \tag{4}$$

Similar to its use in style transfer (Mroueh, 2020b), we assume this transformation does not change the content of the representation, and we therefore do not apply an interpolation scheme to the labels $y_i$ and $y_j$ of datapoints $x_i$ and $x_j$. Instead, we use the label $\hat{y}_{ij} = y_i$ for the virtual representation $\hat{z}_{ij}$.

In Verma et al. (2019a) the interpolation coefficient $\lambda$ is sampled from a Beta distribution $\operatorname{Beta}(\alpha, \beta)$, where $\alpha$ and $\beta$ are hyperparameters. In our experiments we set both $\alpha$ and $\beta$ to 1, so that we are effectively sampling $\lambda$ uniformly from the interval $[0, 1)$.

## C EFFECT OF NUMBER OF TEACHER UPDATES

To better distinguish the effect of the number of teacher updates from the number of training steps between each two consecutive teacher updates, in Figure 4, we plot the accuracy as a function of the number of teacher updates with a fixed number of training steps. That means, in this setup, the total number of training steps increases as we increase the number of teacher updates. For GIFT, we observe an increasing trend in the accuracy as the number of teacher update increases on all the benchmarks. However, for iterative self-training, we only see a benefit in increasing the number of teacher updates for datasets with a range of perturbations in the target domain, such as the Translated CIFAR10 datasets, as opposed to Blurred CIFAR10.

Additionally, Figure 5 shows the effect of the number of teacher updates when the total number of training steps is 1000 for a non-gradual version of GIFT, where all the interpolations are presented to the model simultaneously. We observe that, in this case, compared to GIFT, where the value of the interpolation coefficient $\lambda$ gradually increases, having more teacher updates is less beneficial.

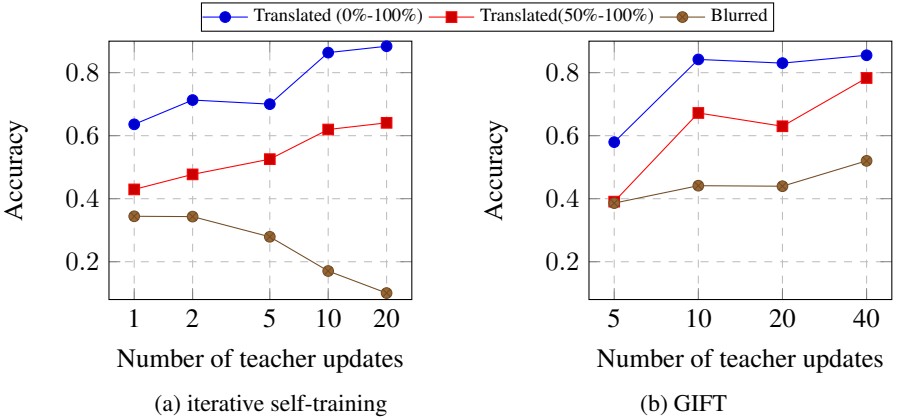

(a) iterative self-training          (b) GIFT

Figure 4: Effect of the number of teacher updates on the accuracy when the number of training steps before each teacher update is fixed and set to 100, for different perturbations of CIFAR10. For Translated (0%-100%) CIFAR10 and Translated (50%-100%) CIFAR10, we see an increasing trend in accuracy as we increase the number of teacher updates for both iterative self-training and GIFT. Whereas for Blurred CIFAR10, the accuracy decreases for iterative self-training.

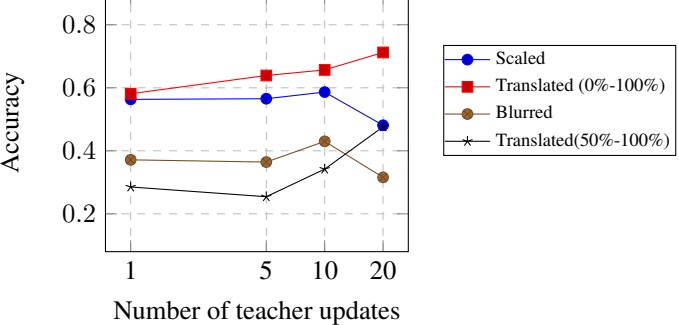

Figure 5: Effect of the number of self-training iterations on accuracy when all interpolations are represented to the model at the same time with the total number of training steps of 1000 for different perturbations of CIFAR10. Similar to iterative self-training, the performance improves by increasing the number of self-training iterations up to a threshold. Beyond the threshold, the performance deteriorates.

