# OpenReview forum: "Gradual Domain Adaptation in the Wild: When Intermediate Distributions are Absent"
_ICLR.cc/2022/Conference — ICLR 2022 Submitted_

### Official Review · Reviewer_KPWG · 2021-10-31

**Correctness:** 4
**Technical Novelty And Significance:** 2
**Empirical Novelty And Significance:** 4
**Recommendation:** 6
**Confidence:** 4

**Main Review:**

Strength:

This paper is well written and easy to understand. The proposed method is simple but effective. The analyses of how, why, and when iterative self-training is helpful can benefit the community, which is important.

Weakness:
1.	Generating intermediate data in domain adaptation is not a new thing, it is recommended to discuss with related works, such as [1,2].
2.	As shown in Table 3, the proposed method does not bring significant improvement over iterative self-training. It is recommended to perform evaluations on more datasets with natural distribution shifts.

[1] DLOW: Domain Flow for Adaptation and Generalization

[2] Unsupervised Adaptation Across Domain Shifts by Generating Intermediate Data Representations


**Summary Of The Paper:**

This paper introduces how to deal with the situation when intermediate distributions are unavailable. The basic idea is to create virtual samples by interpolating source and target representations. The effectiveness of the proposed method is evaluated and results in several interesting conclusions.

**Summary Of The Review:**

My major concern is the effectiveness of the proposed method on datasets with natural distribution shifts. Although this method works well on synthetic data, its applicability to real datasets should be further justified and evaluated.

---

> ### Author Response · Authors · 2021-11-22
> **Response to Reviewer KPWG**
>
> Dear Reviewer KPWG,
>
> Thank you for your feedback and comments. We really appreciate that you find the analysis of how, why, and when iterative self-training is helpful to be beneficial for the community. This is indeed the most important aspect of this paper.
>
> Thank you for pointing out the relevant papers on this topic.  We have extended our Introduction and related work section to include a brief comparison of these methods with GIFT.
>
> Regarding the results of GIFT on natural datasets, we would like to point out that the benefit of using GIFT will kick in when the GAP between the source and target is due to a gradual shift where learning domain-invariant representations is not helpful. The benchmarks from table 3, don’t necessarily have such property, while this is a  common case in real-world problems like concept drift over time. We are excited about this direction and hope that soon we will have benchmarks that showcase the ability of GIFT in a more clear way. In any case, we included experiments on natural images to show that GIFT is still competitive to other domain adaptation methods, even in a setup that is designed for assessing different aspects.

---

> > ### Comment · Reviewer_KPWG · 2021-11-30
> > **Thanks for the response**
> >
> > I appreciate the authors’ efforts in answering my questions. I keep my score.

---

### Official Review · Reviewer_peK3 · 2021-11-01

**Correctness:** 3
**Technical Novelty And Significance:** 1
**Empirical Novelty And Significance:** 1
**Recommendation:** 3
**Confidence:** 5

**Details Of Ethics Concerns:**

No.

**Main Review:**

Strengths:
1. This paper is well-motivated since the gradual domain adaptation is still an under-explored problem.
2. The motivation and the overall idea of the paper is easy to follow.

Weaknesses:
1. novelty concerns:

(a) The main concern for this paper is the novelty. The main idea to address Gradual Domain Adaptation in this paper is to create synthetic data from source and target domains. There are already several previous papers working on this idea, such as DLOW [1], GVB [2], FixBi [3]. Although the above methods are not proposed to directly solve the gradual domain adaptation problem, they can be adapted to this problem. To prove that the proposed GIFT is better, the above methods should be compared after applying to the same problem.

References:

[1] Rui Gong, et al. "DLOW: Domain Flow for Adaptation and Generalization", CVPR, 2019.

[2] Shuhao Cui, et al. "Gradually Vanishing Bridge for Adversarial Domain Adaptation", CVPR, 2020.

[3] Jaemin Na, et al. "FixBi: Bridging Domain Spaces for Unsupervised Domain Adaptation", CVPR, 2021.

2. technical detail concerns:

(a) lambda scheduler: It should be critical to find the best way to transfer the intermediate data smoothly from source to target. However, the proposed lambda scheduler is a linear function with a fixed step size. The step size could be different when facing different datasets, making the current scheduler heuristic and not generalized.

(b) alignment method: it is still unclear which method is adopted for the final model according to the context. Pseudo-random alignment or cost-based alignment? This part is confusing.

3. experiment concerns:

(a) Table 1: The number of self-training iterations is different for iterative self-training and GIFT (5 and 20). This would make the comparison unfair. Why not keep the same training setup?

(b) Separating Tables 1 and 2 is confusing: It is unclear what the difference is between these two Tables. Is the main difference in the total number of training steps (1000 vs. 500)? Moreover, it is not clear why we can claim GIFT is more “robust” than iterative self-training. For Translated (50%-100%) CIFAR-10, iterative self-training raises the accuracy from 0.477 to 0.658 when increasing the training step number from 500 to 1000. However, GIFT drops the accuracy from 0.832 to 0.729. In this case, it is weird to claim GIFT is more robust.

(c) Figure 3: It is weird to compare with different numbers of teacher updates. It would be great to provide an explanation about this.

(d) Table 3: With the same baseline (i.e., Best (A, B, C, D)), GIFT does not have a clear improvement over iterative self-training.

4. other concerns:

(a) The publications in the Reference Section should be updated:

(1) Dan Hendrycks, Steven Basart, Norman Mu, Saurav Kadavath, Frank Wang, Evan Dorundo, Rahul Desai, Tyler Zhu, Samyak Parajuli, Mike Guo, Dawn Song, Jacob Steinhardt, and Justin Gilmer. “The many faces of robustness: A critical analysis of out-of-distribution generalization”. ICCV, 2021.

(2) Ananya Kumar, Tengyu Ma, and Percy Liang. “Understanding self-training for gradual domain adaptation”. ICML, 2020.

(3) Behnam Neyshabur, Hanie Sedghi, and Chiyuan Zhang. “What is being transferred in transfer learning?”. NeurIPS, 2020.

(4) Pang Wei Koh, Shiori Sagawa, Henrik Marklund, Sang Michael Xie, Marvin Zhang, Akshay Balsubramani, Weihua Hu, Michihiro Yasunaga, Richard Lanas Phillips, Irena Gao, Tony Lee, Etienne David, Ian Stavness, Wei Guo, Berton A. Earnshaw, Imran S. Haque, Sara Beery, Jure Leskovec, Anshul Kundaje, Emma Pierson, Sergey Levine, Chelsea Finn, and Percy Liang. “WILDS: A Benchmark of in-the-Wild Distribution Shifts”. ICML, 2021.

(5) Qizhe Xie, Eduard Hovy, Minh-Thang Luong, and Quoc V Le. “Self-training with Noisy Student improves ImageNet classification”. CVPR, 2020.

(6) Barret Zoph, Golnaz Ghiasi, Tsung-Yi Lin, Yin Cui, Hanxiao Liu, Ekin D. Cubuk, and Quoc V. Le. “Rethinking Pre-training and Self-training”, NeurIPS, 2020.



**Summary Of The Paper:**

The paper proposes to address the gradual domain adaptation problem by creating virtual samples from intermediate distributions by interpolating representations of examples from source and target domains.

**Summary Of The Review:**

The proposed method in this paper is not novel although creating virtual examples is a reasonable direction for addressing gradual domain adaptation. I suggest the authors can still work on this direction but propose a more novel way to create virtual examples that contain better intermediate information. The experimental results do not support what the paper claims and do not show significant improvement over previous methods. Therefore, I recommend rejecting this paper.

---

> ### Author Response · Authors · 2021-11-22
> **Response to Reviewer peK3**
>
> Dear reviewer peK3,
>
> Thank you very much for your review. We will incorporate your feedback to further improve the  paper.
>
> We realized that a main concern is the novelty of the proposed method GIFT. First of all, we would like to mention that our emphasis in this paper is more on the side of exploring the power of self-training for domain adaptation and it’s not limited to proposing GIFT. We establish the discussion on the cases where the gap between source and target domain is due to a gradual shift, and where learning domain invariant representations, as proposed by many domain adaptation research works falls short. We explore self-training and show when it works and when it doesn’t and propose GIFT as a solution to remedy lack of intermediate representations in gradual self-training. We believe this paper brings interesting insight into the discussion of domain adaptation and provide practical tips for when and how self-training can be applied  to deal with domain shift.
>
> Regarding comparing GIFT with existing  methods, we have added the research works you’ve pointed out to our paper as related work and discuss how they are related and how they are different from what we discuss in our paper and the setup we consider.
>
> ### Regarding  your concerns about  technical details:
>
> **The lambda scheduler:** We agree that this is not the most adaptive scheduling strategy. The current scheduler we use in the experiment is a simple alternative and there is definitely more room to come up with more elegant/complicated schedulers.
>
> **The alignment methods:** As stated in the paper, we report the result from cost-based alignment (using optimal transport) but our observation was that pseudo-random alignment and cost-based alignment lead to more or less similar results and in practice, we could use the pseudo-random alignments which is simpler and more efficient.
>
> ### Regarding your concerns about experiments:
>
> (a) In table 1, for each method, we have reported the result from  the optimum setup for that method. In Figure 3, We have shown the performance of both methods for different number of teacher updates.
>
> (b) We have placed Table 1 and  2 next to each other to make the comparison easier. Also regarding your question about the statement on the robustness of GIFT compared  to iterative self-training, we  see that when decreasing the number of training iterations, in all case the performance of  iterative self-training method drops, whereas for GIFT, the drop in the  performance is much smaller compared to iterative self-training, or even we see slight improvement on GIFT’s performance when decreasing number of iteration. If we look  at this  phenomena from the opposite direction (the impact of increasing the number of training  iterations)  the statement would not  hold in case  of Translated CIFAR-10.
>
> (c)  In Figure 3, we are trying to illustrate the sensitivity of iterative self-training  and GIFT  to the number of  teacher updates.
>
> (d) Regarding the results of applying GIFT and iterative self-training on natural datasets, we would like to point out that the benefit of using GIFT will kick in when the GAP between the source and target is due to a gradual shift  where  learning domain invariant representations is not helpful. The benchmarks from table 3, don’t necessarily have such property, while this is a  common case in real world problems, like concept drift over time. We hope that soon we will have benchmarks that showcase the ability of GIFT in a more clear way.
>
>
> We hope our responses and the updates to the paper are considered by the reviewer and happy to discuss if there are more comments.

---

> > ### Comment · Reviewer_peK3 · 2021-11-22
> > **Thank you for the response**
> >
> > 1. The authors claim that the main contribution is that "The paper explores self-training and show when it works and when it doesn’t and proposes GIFT as a solution to remedy lack of intermediate representations in gradual self-training." However, self-training is not a new method and the proposed GIFT is not a proper solution. All the components in GIFT are not fully explored (e.g., alignment method, lambda scheduler, etc.)
> > 2. It is still not clear how to demonstrate that GIFT is more robust than iterative self-training by Tables 1 and 2. There is no clear trend in Tables 1 and 2.
> > 3. The comparison of teacher update numbers between iterative self-training and GIFT is not fair since different numbers are used for comparison.
> > 4. The paper claim that Table 3 does not show the obvious improvement mainly because of the dataset used in the paper. Therefore, it would be great to evaluate GIFT using better benchmarks in the future and the results are encouraged to submitted to the future venue.

---

### Official Review · Reviewer_DHFk · 2021-11-01

**Correctness:** 4
**Technical Novelty And Significance:** 2
**Empirical Novelty And Significance:** 2
**Recommendation:** 5
**Confidence:** 3

**Main Review:**

1. This work seems to have certain novelty. I haven't seen papers that anneal the mixup weights. However, it's unclear to me how much this strategy helps DA. I didn't find an ablation study that use a fixed mixup weight.
2. The technical novelty is a bit limited, as it's a simple extension of manifold mixup. To convince readers of its practical value, the authors should do more extensive experiments as well as ablation studies. In natural images, the two used datasets FMoW and Camelyon17 are not typically used to evaluate DA. The authors also should compare with many more existing methods. One important related method is Tent [1].

[1] Tent: Fully Test-time Adaptation by Entropy Minimization, ICLR 2021.


**Summary Of The Paper:**

This work adddresses domain adaptation (DA) by a GIFT method. GIFT consists of a manifold mixup technique, which generates virtual samples by mixing up the features of source and target domains samples. The mixup coefficient is annealed over time to bias towards the target domain. Another ingredient of GIFT is a co-teaching strategy that lets two networks teach each other. On a few small synthetic as well as natural image datasets, GIFT outperforms a few baseline methods.

**Summary Of The Review:**

The technical novelty of GIFT is limited. Hence, the authors should focus on showing the empirical benefits of GIFT. However, the experiments are highly insufficient, with missing ablations and baselines. Moreover, the evaluation datasets are not so popularly used. Therefore, it's hard to judge how much empirical value this work brings.

---

> ### Author Response · Authors · 2021-11-22
> **Response to Reviewer DHFk**
>
> Dear  reviewer  DHFk,
>
> Thanks a lot for your time for reviewing the paper.
>
> It seems the main concern is the novelty of our work. We would like to mention that our paper does not extend manifold mixup, but incorporates the idea to resolve the issue with gradual self-training when intermediate steps do not exist. We believe our work brings new practical insights on the topic of domain adaptation. Note that in our paper, besides proposing GIFT, we point out settings and scenarios where simply learning domain invariant representations does not solve the domain adaptation problem (Figure 1 illustrates this with a toy example). Furthermore,  we investigate when and how iterative self-training can be beneficial for gradual domain adaptation, and we show that it fails when there is a large gap between the two distributions.  We show that while iterative self-training with confidence thresholding works well in condition (i) even though the annotations for the degree of the shift are missing, it does fail in condition (ii) when the intermediate samples are completely missing and there is a large gap between source and target domains. We relate the idea of gradual self-training to curriculum learning and show that under condition (i), self-training incorporates an implicit curriculum based on the teacher’s confidence score.
>
> Regarding including ablation experiments to compare the current approach with the setup where we use a fixed mixup rate: We do have an ablation experiment that shows the impact of the number of steps when annealing $\lambda$ (the mixup coefficient) (Figure 3 and 4).  The case where the number of steps is set to 1, basically means we only use a mixup ratio of 0.5 (as well as 0.0 at the end). We will add data points with the number of steps = 1 and 2 in the camera-ready version of the paper.
>
> About extending the experiments to include more natural datasets, we would like to point out that,  in this paper, we target cases where the type of shift is such that simply learning domain invariant representation won’t be sufficient, and we would  need benchmarks in which we do have access to the intermediate steps of the shift so that we can evaluate the effectiveness of the virtual interpolation. This is an interesting and new research direction and there is still a lot of work to be done, including designing and creating suitable benchmarks.
>
> We have updated the paper to address the minor comments and add missing references. We hope that our response and the updates in the paper are considered by the reviewer.  We are happy to address any additional comments.

---

### Official Review · Reviewer_Les7 · 2021-11-02

**Correctness:** 2
**Technical Novelty And Significance:** 3
**Empirical Novelty And Significance:** 2
**Recommendation:** 3
**Confidence:** 4

**Main Review:**

This work considers the problem of gradual domain adaptation, where a model should be gradually adapted from the source to the target domain. I found the idea of generating data (at a lower-dimensional space) from intermediate distributions interesting and appealing from a practical perspective. In the following, I present my major concerns, questions, and suggestions.


- It is not clear from the manuscript why gradual domain adaptation is a setting that should be considered in cases where no samples from intermediate distributions are available. As far as I understand, gradual domain adaptation is a setting to be considered when the distribution yielding the data is evolving over time (Kumar et al. 2020) and the goal is to incorporate this knowledge about the problem structure in the model adaptation process. Given that, it is not clear to me why considering this setting in a case where the data distribution is not gradually shifting would make sense. If this is the case, why not directly use any other domain adaptation approach?


 - A major claim of this contribution is that by mixing-up representations of both source and target domains, examples from “intermediate” distribution would be generated. Although I can get a rough intuition of what an intermediate distribution could be (e.g. a mixture of source and target domains), there is no definition or discussion regarding this in the manuscript. I suggest the authors include in the manuscript a clear definition of what such distributions are, as well as include evidence that manifold mix-up is capable of generating samples from them.


- Experiments
  - The authors mentioned that the decrease in confidence/accuracy observed on examples with higher levels of perturbation observed in
Figure 2 confirms that the reason behind the success of iterative self-training is the implicit curriculum strategy. Although I understand that this observation indicates the existence of an implicit curriculum, it is not clear to me why it indicates it is the reason behind the success of iterative self-training.
  - It is not clear to me whether the results presented in Tables 1 and 2 are good. Even though I understand that the goal of this work isn’t to achieve state-of-the-art results in benchmarks, it is hard to assess the merit of the improvements reported in these tables without knowing how established baselines would perform in such test cases. I strongly encourage the authors to include for comparison at least DANN and CDAN.
  - In Table 3, the authors presented results obtained on two datasets. They mentioned that in the case of Camelyon17, hospitals 0, 1, and 2 were considered as source domains while hospital 3 was the target. However, in Table 3, it seems that only results with domain 0 as the source were reported. Why is it the case? Please clarify this and modify the text accordingly.
   - It is hard to tell whether the results presented in Table 3 indicate a relevant improvement of self-training approaches and the proposed GIFT in comparison to the considered baselines. The authors did not report if more than one run was performed. In case only one run was performed, I don’t think it is possible to draw conclusions from these experiments since models adapted via DANN, for example, are known to be quite sensitive to the initialization.


- In addition to the aforementioned concerns, this manuscript presents several presentation and clarity issues that make it difficult to understand. In the following, I outline the major issues:
  - In Section 2, the authors introduced the considered setting but there are several missing points regarding the introduced notation. Please properly define $n_s$, $d$, $k$, and $n_t$. Moreover, the authors did not specify the underlying assumptions of the proposed approach, i.e., is the covariate assumption required? Is label shift allowed?
  - Furthermore, in Section 2, the authors mentioned that “The goal is to bridge the domain difference and learn a good classifier for the target domain.” What exactly does “a good classifier” mean in this sentence? I believe the authors are considering the risk minimization setting from Ben-David et al. 2010, but please let this clear in the text.
  - It is quite difficult to parse the information contained in Algorithms 1 and 2. Several variables were not introduced, and there is no comment to explain what each line is doing. Summing this up with the fact that there is barely any explanation about GIFT training procedure and the Label-based Random Alignment throughout the text, it is hard to properly understand what the main contributions of this work are really doing. Moreover, the clarity of the Algorithms and the aforementioned contributions should be improved to facilitate the reproduction of the reported results.


- Minor
  - It is hard to compare the results reported in Tables 1 and 2 because they are placed too far apart in the manuscript. I think it is possible to merge both into a single table.
  - The symbols $P_s$ and $P_t$ are used to denote different objects in the text. In Section 2, they are referred to as the source and target domains, respectively, while in Algorithm 1 they denote source and target datasets, respectively.


**Summary Of The Paper:**

This work proposes an iterative self-training approach for unsupervised domain adaptation. In particular, the authors aim at gradually adapting a model trained on a source domain to the target domain. Based on the claim that previous work on this setting assumed that samples from distributions that represent gradual changes from the source to the target domain are available when adapting the model, the authors proposed a strategy to generate such intermediate samples for cases where they are not available. The introduced approach, named GIFT, consists in performing manifold mix-up between representations of examples from the source and target domains considering an increasing value of the hyperparameter that accounts for the weight of the representation of the target domain example. By doing so, the authors claim an automatic curriculum is introduced in the training. Moreover, since labels from the target domain examples are not available, the authors also introduced a heuristic to pair the examples that are mixed. GIFT was empirically validated on experiments with synthetic domain shifts on the CIFAR-10 dataset and was shown to outperform iterative self-training in terms of target accuracy. Experiments on two datasets presenting natural domain shifts were also performed.

**Summary Of The Review:**

This work proposes an iterative self-training approach for unsupervised domain adaptation. In particular, the authors aim at gradually adapting a model trained on a source domain to the target domain. The main contribution is a strategy to generate examples from the so-called intermediate distribution between the source and target domains. Despite the reported empirical improvement over the considered baselines, I found the motivation of the proposed approach unclear, and I found that the main claim is not well-supported (i.e. is GIFT indeed capable of generating examples from intermediate distributions? Also, what are such intermediate distributions?). Moreover, the manuscript lacks clarity in several aspects and I have concerns regarding the significance of the reported results since it seems only a single run was considered in each test case. All in all, my initial assessment is that this manuscript is not ready for publication yet.

---

> ### Author Response · Authors · 2021-11-22
> **Response to Reviewer Les7 (Part  I)**
>
> Dear reviewer Les7,
>
> Thanks a lot for the substantive and detailed review and for the insightful comments. We really appreciate your time and feedback and have incorporated them to improve the paper.
>
> Hereby we address each of your comments separately:
>
> **Why gradual domain adaptation when there exist no intermediate samples?**
>
> TL;DR:
> Gradual domain adaptation is based on assumptions that perfectly fit the cases where the gap between source and target domain is due to a gradual shift, like concept drift over time. Such inductive biases in GIFT help it to be effective in these cases compared to general domain adaptation methods (in particular those that are based on learning domain invariant representations).
> Detailed response:
>
> To clarify the main motivation behind this work, we would like to emphasize that the contributions of our paper are two folds:
> Our first contribution in this paper is to comprehensively explore and investigate if and how self-training can be applied as a domain adaptation technique, in cases where (i) the intermediate samples exist, but they are not annotated with the degree of shift. (ii) the intermediate samples do not exist.
> Setup (i) is the generalization of the setup discussed in [1], and we show that in this case, self-training can still be beneficial. We agree with the reviewer that setup (ii) is basically the standard domain adaptation setup, but we argue that it is still interesting to show how self-training performs in these settings: The common approach in many other domain adaptation techniques, is to learn “domain invariant representation”, whereas for some types of shift, e.g., if there is concept drift, the problem is not solvable by learning more generalizable representations. This is the main message that we try to show in Figure 1. In this case, self-training-based methods sound the most intuitive approach to shift the model, but we show that they won’t work if there is a large gap.
> Our second contribution is to propose GIFT, to leverage interpolations of features to fill in the gap between the source and target domain and improve the efficiency of self-training to shift the model. Again the focus of GIFT is **not** to learn domain invariant representation, but to shift the model toward the target domain distribution, which is necessary in cases where the source and target domain are not consistent.
> Additionally, in general, the assumption that changes in the data distribution, naturally occur gradually, could be helpful as a kind of inductive bias, that even in the absence of samples from intermediate steps of the shift could help the model adapt in a more data-efficient manner.
>
>
> **What an intermediate distribution could be?**
>
> We thank the reviewer for bringing this up. This is indeed a very valid point to discuss. Since Mixup and Manifold Mixup are well-established augmentation methods in supervised settings (where the ground truth labels for both mixed examples are known), in this paper we rely on the existing work on this topic that shows the usefulness of such interpolations [2,3]. Moreover, as another piece of evidence to show that this is a promising direction, we can refer to recent works in the context of style transfer, where they apply similar approaches to generate intermediate examples, from the interpolations of features [4]. We briefly discuss this in the updated paper with pointers to the works that originally established the concept of mixup representations.
>
> **Concerns regarding the experiments:**
>
> Regarding the implicit curriculum helping iterative self-training: Besides the general benefits of curriculum learning to improve the learning process, here our speculation is mainly based on the comparison of these two setups: self-training with and without the smooth curriculum. When the intermediate examples are missing, the model can not form a smooth curriculum and fails to perform well, while using the intermediate examples and therefore forming a smooth curriculum, self-training is effective. This is based on the comparison of the results of translated CIFAR-10 with gap, with translated CIFAR-10 without the gap in Table 1, rows 3 and 4.
> Thanks for the comment on datasets used for the experiments in Table 3. We used the three first domains as a training set, during fine-tuning, but for adaptation only used labeled data from the first domain. We updated the paper to clarify this.
>
>
> ### References:
> 1. Ananya Kumar, Tengyu Ma, and Percy Liang. Understanding self-training for gradual domain adaptation. arXiv preprint arXiv:2002.11361, 2020.
> 2. Zhang, Hongyi, et al. "mixup: Beyond empirical risk minimization." arXiv preprint arXiv:1710.09412 (2017).
> 3. Verma, Vikas, et al. "Manifold mixup: Better representations by interpolating hidden states." International Conference on Machine Learning. PMLR, 2019.
> 4. Mroueh, Youssef. "Wasserstein style transfer." arXiv preprint arXiv:1905.12828 (2019).

---

> > ### Author Response · Authors · 2021-11-22
> > **Response to Reviewer Les7 (Part II)**
> >
> > Thanks also for additional comments on minor issues with notation, the definition of a “good” classifier, free text explanation of algorithm 1 and 2 (this is in Section 3), adding explanation/definition for notations used in algorithms, and the place of Table 1 and Table 2. We have updated the paper to address all these comments. We are also currently working on adding DANN to table 1 as well as reporting mean and standard deviation in Table 3 for the camera-ready version.
> >
> > We hope that the reviewer considers our responses and the updates to the paper and reconsiders the assessment scores. We are happy to incorporate additional comments.

---

> > > ### Comment · Reviewer_Les7 · 2021-11-29
> > > **Further comments**
> > >
> > > Dear authors, thank you for your response and updates on the paper. Although the authors addressed some of the points I raised in my review, the most important concerns I have regarding this work still remain. For example, it is not clear to me what the authors consider as an intermediate distribution (i.e. the paper lacks a proper *definition* of intermediate distribution). Moreover, it is unclear if mixup-based approaches are capable of generating samples from such distributions. Notice that I'm not concerned about the merits of mixup as an approach for *data augmentation* as this is already very clear from previous work.
> > >
> > > Given that my major concerns were not addressed by the rebuttal, I decided to keep my score.

---

### Official Review · Reviewer_v3Ep · 2021-11-02

**Correctness:** 3
**Technical Novelty And Significance:** 3
**Empirical Novelty And Significance:** 3
**Recommendation:** 5
**Confidence:** 2

**Main Review:**

+ The problem is well motivated and illustrated through different experiments and toy examples.
+ paper is well written and easy to follow

Clarification need:
- In Algorithm 2: matching scores are 1 for the same class, so if there are false positives(which happens a lot in real datasets), then the alignment will be biased towards them leading to poorer student network training?
- the choice of target examples to be mixed may create difficult examples early on, it is not clear how random assignments can overcome this situation? Also, the Align function along with the \lambda value will together decide, the hardness of the intermediate examples created.
- lambda_scheduler tuning is not discussed properly, will it depend on the gap between the two domains? if yes, then how should be handled?
- mismatch in lambda_scheduler step size notation \delta in Algorithm 1 and \sigma on page 3
- figure 1 has "gradual self-training" as a caption in one of the subplots but this method is not mentioned or cited.


**Summary Of The Paper:**

This work builds on the previous work[1] and extends it for a more general scenario, where per sample intermediate distribution shift is hard to quantify. They propose gradual feature interpolation for the case where samples from intermediate distribution are missing. Iterative self-training fails in such cases. The work presents results on synthetic and natural distributions to evaluate their claim.

[1] Ananya Kumar, Tengyu Ma, and Percy Liang. Understanding self-training for gradual domain adaptation. arXiv preprint arXiv:2002.11361, 2020.

**Summary Of The Review:**

The work puts forwards an interesting work to gradually align two domains. Though different experiments are shown to support their claim, there are few clarifications needed as mentioned in the main review.

---

> ### Author Response · Authors · 2021-11-22
> **Response to Reviewer v3Ep**
>
> Thank you very much for your thoughtful comments and feedback.
>
> Here, we address the clarity issues that you have raised (we have updated the paper accordingly):
>
> 1. In response to your comment about alignment bias in algorithm 2, this is indeed correct, however, this is generally the main challenge with self-training and using pseudo labels, not specific to our paper. The assumption here, which to some extent helps us to escape from such issue is that by filtering the examples based on the confidence of the model, we reduce the chance of using examples that are hard to classify for the teacher model and are falsely labeled, and also that since we take smaller steps, it is easier for the model to correct itself as the training proceeds (Figure 2, illustrates that this is indeed happening for the Translated CIFAR-10 case). Note that we can go further and try to reduce this effect by adding a degree of noise or by applying an optimal transport based approach for the alignment and using other features, such as the distance of representations, which by itself deserves a comprehensive study.
>
> 2. Regarding your question on the difficulty of generated virtual intermediate samples at the early stages of training, we have two mechanisms to account for such phenomena:
> (1) annealing the value of \lambda so that in the earlier steps, the intermediate representations are more similar to the representations of source samples;
> (2) filtering samples based on the confidence of the teacher on the interpolated generated samples. The idea here is that, since the interpolated representation moves away from the representations of the source examples, the prediction of the teacher model and its confidence value, are more reliable, compared to their prediction of the actual target examples.
>
> 3. Regarding your question about tuning the lambda scheduler: That is true that the scheduling of the lambda can have a significant impact on the performance of the models. In our experiments, we study this question. We employ a linear decay (from lambda=1 to lambda=0), where the parameter step_size determines the number of intermediate steps or the number of teacher updates. Additionally, there is another hyper-parameter, which is the number of training iterations in each intermediate step. We run a series of ablation studies (Figure 3 and Figure 4 in the appendix) to show the impact of these two hyper-parameters and it seems both the type of shift and degree of shift (which represent the gap between source and target domains) can impact the optimum values for them.
>
> 4. Thanks for pointing out the inconsistency in notation, we have fixed this in the new revision of the paper.
>
> 5. Thanks for pointing this out, we have added the reference in the caption of Figure 1.
>
> Once again, thanks for the comments. We sincerely hope that we have addressed your concerns in our rebuttal. Considering that you find the problem well-motivated and our paper interesting and well written, we appreciate reconsideration of the assessment scores if there are no additional issues. We are happy to discuss and address if there are any additional comments.

---

> > ### Comment · Reviewer_v3Ep · 2021-11-26
> > **Response to Author**
> >
> > Thank you for answering my questions. I agree with experimentations done with synthetic datasets to show the limits of the self-training methods and they are interesting. Having said that the concerns regarding GIFT formulation(alignment) still exist, which is also a significant contribution of the paper.

---

### Author Response · Authors · 2021-11-22
**General Response**

We thank all the reviewers for their time and effort in reviewing our paper. There are many valuable and interesting insights in the comments and feedback.

Since most of the comments from the reviewers are focused on GIFT, we emphasize that one of the most important contributions and a major part of our paper, which could be overlooked, is to study the power of self-training and iterative self-training to deal with distribution shifts, and investigate when it works and when it fails.

To do so, we start with an example in Figure 1 and showcase a setup where learning domain invariant features (which is the foundation stone of many domain adaptation methods) can not solve the adaptation problem, but we can still benefit from techniques such as gradual self-training [1]. Next, we design and create a dataset with various synthetic shifts and examine the performance of self-training and iterative self-training under two scenarios:
(i) intermediate steps exist but are not annotated with the degree of shift.
(ii) intermediate steps do not exist at all and there is a significant gap between source and target distributions.
We demonstrated that under condition (i) iterative self-training with confidence thresholding, forms and implicit curriculum which helps the model to adapt to the target domain gradually. However, under condition (ii) iterative self-training fails.
These studies already present the reader with useful information on self-training and gives intuition on their advantages and disadvantages.

Besides the above, we propose GIFT, the method to improve the performance of iterative self-training when samples from intermediate distributions are completely missing and there is a large gap between source and target distribution. In fact, GIFT is a preliminary solution, to mitigate the issue by generating samples from virtual intermediate distributions. In designing GIFT we do explore existing ideas to employ feature interpolates as a proxy to representations of examples from intermediate distributions. But our experiments with GIFT are more explorative and there is a lot of room for improvement and hopefully, more advanced methods can lead to even more reliable results, given the perspectives that GIFT offers. We believe our paper leads the way toward a lot of interesting research in this direction.

We are excited to see self-training becoming a solid technique and believe a study like ours, which carefully explores self-training under controlled experiments improves our understanding of it and helps us all to make better progress in this direction.

We have updated the paper to incorporate comments from reviewers and here we provide responses to their individual comments. We are happy to discuss more and incorporate any additional comments and suggestions.

---

### Decision · Program_Chairs · 2022-01-20

**Decision:**

Reject

**Comment:**

Most reviewers agree that the paper addresses a relevant problem. However, they also  believe that
the paper lacks in several points : not-well supported claim, sometimes clarity, incremental in term of novelty.